# A Study on the Preparation and Performance of Ultrafine Powder Made of Industrial Hemp Degumming Residue

**DOI:** 10.3390/polym16243473

**Published:** 2024-12-12

**Authors:** Sarker Md Shamim, Yonghe Huan, Linli Gan, Shangyong Zhang

**Affiliations:** 1School of Textile Science and Engineering, Wuhan Textile University, No. 1 Sunshine Avenue, Jiangxia District, Wuhan 430200, China; ssharker525@gmail.com (S.M.S.); yonghehuan2023@163.com (Y.H.); llgan@wtu.edu.cn (L.G.); 2State Key Laboratory of New Textile Materials and Advanced Processing Technologies, No. 1 Sunshine Avenue, Jiangxia District, Wuhan 430200, China

**Keywords:** residue hemp fiber recycling process, chemical treatment with cellulose, making an ultrafine powder, average particle size

## Abstract

Industrial hemp, one of the most widely available and extensively produced varieties, generates a substantial amount of waste in the form of hemp cellulose. This study uses a recycling method combining crushing and acid treatment to convert leftover hemp fiber into ultrafine powder. A scanning electron microscope (SEM), an atomic force microscope (AFM), Fourier transform infra-red spectroscopy (FTIR), and X-ray diffraction (XRD) were used to examine the morphology of acid-treated hemp fiber heated to 200 °C and crushed into powder. The decrease in intensity, fiber surface crystalline, and grain size was analyzed. It became apparent that fiber strength decreased, and fiber roughness significantly increased after acid treatment. The degree of crystallinity of the broken fibers decreased significantly. The proposed method was a simple and effective method for converting leftover hemp fiber into ultrafine powder. In approximately 3 to 5 min, about 1 kg of dry ultrafine powder with a particle size of 38.68 μm was produced. This production method will significantly enhance future industrial applications of hemp residue.

## 1. Introduction

Worldwide hemp production is in the range of 32,140 ha to 142,883 tons, resulting in an average yield of 4.45 t ha^−1^ [1]. Extensive regional cultivation ranges and various uses have made hemp a significant global resource. The top three countries by production area for hemp, according to FAO Stat (2018), are North Korea (21,247 ha), France (12,900 ha), and Canada (555,853 ha) [2]. In 2019, the industrial hemp market was expected to be valued at USD 4.71 billion worldwide [3]. The three biggest markets for hemp are food and beverage, fiber, and paper; these sectors represent a billion-dollar worldwide industry [2]. China (Asia), Canada, the USA, and Europe contributed roughly 29.1%, 35.0%, 6.6%, and 17.9% of global hemp production in 2022, according to data issued by the EIHA [4]. More than 30 countries are currently involved in the global hemp trade, with nearly 651 tons falling within the legal framework for cannabis and textile consumption. This rapid product growth generates millions of tons of textile waste annually [5]. Studying the percentage of short fibers in hemp, often removed during the production of hemp yarn and fabric, offers insights into its waste. Biomass stalks and fibrous material are the main waste types connected to industrial hemp degumming and processing. Hemp fiber contains a short-fiber content of 7.72% [6], with hemp stalks yielding 40% and creating 60% waste. The total waste from opening to spinning can range from 8.5% to 18%, leading to considerable destruction of natural resources and financial losses. Additionally, environmental issues such as water and soil pollution arise from these circumstances [7,8]. As a result, these challenges must be addressed effectively and efficiently. Consumers worldwide are becoming increasingly concerned about this situation. Utilizing residual cellulose polysaccharide fiber to create valuable products could offer the textile industry enormous revenue potential, should these waste fibers be recycled [9].

Hemp waste can be recycled in several ways, the two main methods being chemical and physical processes. Since the mid-20th century, particle technology has shown that turning materials into ultrafine powders affects their energy activity and surface area [10]. These modifications have prompted the creation of specialized processing equipment. After being processed into ultrafine powder, materials display certain modifications, including changes in surface area, energy activity, and interface characteristics [11,12]. Gaining an in-depth knowledge of ultrafine particles and their properties has been crucial in developing processing equipment. Due to its altered surface characteristics, ultrafine hemp powder is particularly intriguing since it can stabilize formulations from food to cosmetics. The process makes it possible to modify hemp ultrafine powder to have balanced surface amphiphilicity, enabling it to stabilize water-in-oil emulsions, such as those used in food, cosmetics, and clothing [13]. Ultrafine hemp powder is widely utilized across various industries, including composite materials, chemicals, biology, catalysis, and medicine, demonstrating its extensive applications [14,15]. Hemp ultrafine powder is used to reinforce composites; hemp fibers offer several advantages over synthetic fibers and can be effectively employed in various applications [16], including biofuels, agricultural products, animal bedding, industrial and automotive plastics, textiles, paper, construction materials, food, beverage, cosmetics, and personal care products. Hemp fibers are versatile, with advantages over synthetic fibers in improving composites. Hemp fiber is primarily composed of cellulose, hemicellulose, lignin, and waxy materials [17]. The macromolecular polysaccharide in hemp cellulose comprises β-1, 4 glycosidic linkages and D-glucose, which is a type of covalent bond. Three components contribute to the high chemical reactivity of cellulose: polysaccharide hydroxyl groups (-OH), which cause a high tendency toward crosslinking and chemical changes [3]. Cellulose is a polysaccharide with a high degree of chemical reactivity, primarily due to its hydroxyl groups. However, cellulose’s strong chemical bonds, crystallinity, and high polymerization levels make it challenging to convert hemp fiber residues into ultrafine powders. Several methods can be used to make cellulose powders: mechanical and chemical [18]. Both chemical and mechanical processes are used to create cellulose ultrafine particles. To create fine particles, the chemical method frequently uses dangerous and complicated compounds [19,20], while the mechanical crushing method is easier but challenging and ineffective due to the fibers’ flexibility and strength [21]. Hemp fiber is known for its high strength and durability, with a tensile strength of about 550 MPa, higher than that of cotton (287 MPa) and wool (100 MPa) [22]. Few research investigations have focused on the mechanical and chemical milling of wool and silk powder [15,23]. Cotton and wool powder, with particle sizes around 60 μm and 40 μm, respectively, have been produced using freeze-milling and millstone milling techniques. Processing the fibers into dry powder took approximately 1.5 h for 1 kg [9,18].

In this work, hemp fibers were first treated with acid to affect their morphology, making it easier to break them into effective powder. Waste hemp fibers were efficiently converted into micron-sized powders by combining chemical and physical processes without complex procedures. Ultimately, after 3 to 5 min, about 1 kg of dry powder with a particle size of 38.68 μm was produced. This method’s dry ultrafine hemp powder output and efficiency were significantly higher than those of the ball milling, cutting milling, bead milling, millstone, and spray drying techniques. This method holds much potential for future production utilization.

## 2. Experiment

### 2.1. Materials

Waste hemp fiber was obtained from Wuhan Hemp Biological Technology Co., Ltd. (Wuhan, China). Sulfuric acid (H_2_SO_4_) was sourced from Wuhan Jiangbei Chemical Reagent Co., Ltd, (Wuhan, China). The XL-06B mixer and refining agent YK were purchased from Kekai Fine Chemical, Shanghai Co., Ltd., (Shanghai, China). Ethanol absolute (C_2_H_6_O) was purchased from Sinopharm Chemical Reagent Co., Ltd, (Shanghai, China).

### 2.2. Acid Treatment and Modification of Hemp Fiber

First, a beaker containing 200 mL of boiled deionized water was placed in a 100 °C water bath for insulation. Then, 2 g/L sulfuric acid, 1 g/L YK agent (which aids in the dilution of chemicals into fibers), and 10 g of leftover untreated hemp fiber (cut to 2 mm × 3 mm) were sequentially added to the beaker. The bath ratio was 1:20. After heating the beaker for 1 h, the hemp fiber was washed 7–8 times until the pH value reached 7. Next, a padding machine or hand roller was used to remove excess water. Finally, the hemp fiber was placed in an oven and baked at 200 °C for 1 h. Additionally, powder was prepared for AFM analysis by dissolving the hemp powder using C_2_H_6_O at 39.82 g/L with hemp powder at 0.70 g/L for 30 min, Figure 1. The acid modification process is illustrated in Figure 2.

### 2.3. Modified Hemp Fiber on a Crusher

Firstly, the hemp fiber was cut to 2 mm to 3 mm with small scissors before chemical treatment. About 1 kg of modified hemp fiber, with a short length, was used in a small Chinese Medicine crusher (XL-06B, Aashia Company, Shanghai, China). It has a high-quality lid with 3 blades: a crushing blade that helps crush the fiber, a bottom blade for grinding the hemp cellulose, and a cutter head that cuts the yarn into small pieces. The device has a power of 1100 watts, a voltage of 220 volts, a fineness of 200, and a speed of 25,000 revolutions per minute. By running the machine at high capacity, the hemp cellulose broke down. This process took 3 to 5 min to crush around 1 kg of ultrafine hemp powder.

### 2.4. Characterization

The morphology of the hemp fiber, following its coating with gold powder, was examined using a scanning electron microscope (SEM, JSM-7800, Japan Electronics, Tokyo, Japan). An accelerating voltage of 5 kV and a magnification of 5000× were used.

The surface roughness of the hemp fiber and powder was measured before and after acid treatment using an atomic force microscope (AFM, model SPM-9700, Shimadzu, Kyoto, Japan). Hemp powder was converted into a liquid using ethanol (C_2_H_6_O) for 30 min. A scanning area of 5 μm × 5 μm, with a 2 × 2 μm^2^ zoom, was used to measure each sample.

Fourier transform infrared spectroscopy (FTIR; IRT Racer-100; Shimadzu Corporation, Kyoto, Japan) was used to determine the spectral resolution, at 4 cm^−1^. The samples were analyzed in a nitrogen atmosphere using the Total Reflectance (ATR) mode. Forty scans were performed on each sample.

After treatment and milling, the hemp fiber underwent an X-ray diffraction (XRD) study using an X’Pert Pro MPD (PA Analytical, Almelo, The Netherlands) to determine crystalline alterations. The theta testing range was 0–40°, and the scan rate was 5°/min. The degree of crystallinity in a sample was determined using the crystallinity index (CI) [24].
CI% = I200 − IamI200 × 100CI\% = \frac{{I_ {200} − I_{am}}} {I_ {200}} \times 100CI% = I200I200 – I am × 100
where I_{am} is the minimum peak intensity of the amorphous phase at about 17.1°, and I_ {200} is the peak at the 2θ angle intensity of the crystalline phase at about 22.5°.

## 3. Results and Discussion

### 3.1. Changes in the Morphology of Hemp Fiber Before and After Crushing

The SEM images shown in Figure 3a–h depict OHF, MHF, THF, and HP, demonstrating the modified hemp fiber surfaces after acid, temperature, and powder treatments. The process was highly magnified. Figure 3a shows that the original hemp fiber surface was slightly uneven compared to after the acid treatment. Figure 4b shows that, after alkali treatment, the surface became rougher and more breakable. Concentrated sulfuric acid can effectively penetrate the cellulose structure, destroy the orderly accumulation of molecular chains, and break the hydrogen bonds inside cellulose [25]. After treatment with high temperatures of 200 °C and acid, the fiber surfaces became weaker, as shown in Figure 3c. The etched fiber texture differed from the original. On the other hand, in Figure 3e,h the low-magnification picture shows OHF (Figure 3a) appearing to be smoother than MHF (Figure 3e) and THF (Figure 3g). This demonstrates how light scattering at different levels on a uniform fiber surface altered the fiber’s luster [21]. Figure 3d,h shows more imperfections and uneven edges on the crushed powder surface compared to the fiber. Dry hemp fiber material is ground more finely. Generally, as particles become finer, their shape tends to become more rectangular and uniform [26]. Since untreated hemp fiber cannot be ground using the method of this study to make a fine powder, there is no original hemp powder sample for SEM image comparison.

### 3.2. Surface Roughness of Hemp Morphology After Treatment and Crushing

Three-dimensional (3D) AFM images of hemp fibers at various temperatures and in powder form are displayed in Figure 4a–h. The surface features bumps and has an uneven distribution across all samples. For OHF (a–e), the surface roughness measurements were Ra: 270.365 nm, Rz: 1.30 μm, and Rp: 615.901 nm, with a scanning scale of 5 μm × 5 μm. These results can still be used as a reference when comparing various samples on the same scanning scale, as they display layers of unevenness in different models. After chemical treatment, MHF (b–f) displayed Ra, Rz, and Rp values of 109.491 nm, 846.889 μm, and 423.870 nm, respectively, indicating a significant reduction in roughness. This shows that the height variations on the treated fibers were more consistent than those of OHF, THF, and HP. Sulfuric acid can dissolve lignin, hemicellulose, and pectin, among other non-cellulosic fiber components. The cellulose fiber in hemp is hydrolyzed and oxidized by sulfuric acid [27]. The hollow structure of hemp fiber was exposed, and its surface etched. More cellulose molecules were exposed as the fiber’s specific surface area increased when it broke down to a certain extent [27,28]. After treatment with diluted sulfuric acid, the clearance rate of hemicellulose reached 98.05%, with high lignin removal achieved due to specific interactions with p-TsOH hydrotrope, aiding in lignin removal. Pectin was frequently hydrolyzed in various processes using acid treatments [29,30].

This may influence the fibers’ mechanical and physical properties, such as stiffness, strength, fineness, and moisture absorption. Hydrolysis can decrease the fiber diameter of cellulose [31]. For THF (c–g), after treatment at 200 °C, the fiber roughness increased, with Ra, Rz, and Rp measuring 125.854 nm, 1.33 μm, and 686.299 nm, respectively [32]. HP (d–h) showed Ra, Rz, and Rp values of 376.507 nm, 2.692 μm, and 1.483 nm, respectively.

As a result of etching, the degree of surface cracking in the hemp fiber was consistent, and the height of surface protuberances was more uniform than that of OHF. HP had the highest roughness because high-speed operation, using various blade types, caused random shearing and crushing of the powder in different directions [33]. Ensuring uniform crushing across the powder’s surface was challenging [33,34].

### 3.3. Analysis of Strength Loss of Modified Hemp Fiber Using Crusher

Due to significant strength loss and breakability after acid treatment at 200 °C, it was difficult to extract modified hemp fiber using a single-fiber strength tester. This weakness was evident in the SEM images (Figure 3), showing MHF and THF fibers as more fragile. Therefore, both acid-treated and non-acid-treated fibers were combined using the same crusher to create the powder. As shown in Figure 1, after acid treatment and exposure to high temperatures, the original gray hemp fiber turned brown, while the untreated fiber looked as if it were black. The acid treatment process essentially weakened the strength of the hemp fibers, making it easier to pulverize into micron-sized powder.

When attempting to crush raw hemp fiber, it was evident that the original residual hemp fiber was solid and difficult to break down using this method. However, sulfuric acid breaks down cellulose fibers into smaller particles. One of the most common chemical treatments is acid hydrolysis. Sulfuric acid can effectively penetrate the cellulose structure of hemp at high concentrations, disrupting the orderly stacking of molecular chains and breaking hydrogen bonds within the cellulose [35]. Breaking the hydrogen bonds in amorphous cellulose leads to the formation of highly crystalline cellulose nanocrystals [36]. The use of sulfuric acid to obtain nanocellulose essentially breaks the β-1,4 glycosidic bonds between cellulose macromolecules and removes the amorphous regions of cellulose [37].

The H^+^ ions generated by sulfuric acid damage the glycosidic linkages in cellulose macromolecules during acid treatment, especially when heated to 200 °C, as shown in Figure 2. Since glycosidic linkages are acetals, they become unstable when exposed to high temperatures and acids. Acting as a catalyst, the acid lowers the activation energy of glycoside bonds and accelerates the hydrolysis of cellulose. Cavities formed on the fiber’s surface due to hydrolysis, as shown in the SEM images, reduced surface light reflection. Consequently, the thread became less lustrous and significantly weaker [38,39]. The crusher, operating at high speed with three blades, applied pressure and force between soft fibers, resulting in friction and mechanical breakdown, as shown in Figure 1. As a result, the sample was smoothly crushed into fine powder.

## 4. Crystallinity of Hemp Fiber and Powder

Normalized XRD patterns of OHF, MHF, THF, and HP are presented in (Figure 5a). The dispersion peak of OHF is at a maximum of about 22.5°, with a minimum of 16.3°. After sulfuric acid treatment, MHF shows new peaks at 14.8° and 22.5°, which appear much weaker. This demonstrates a modification in the crystalline area. After heating to 200 °C, the THF and crushed hemp powder (HP) graphs show peaks at 14.70° and 22.5°, which are lower compared to OHF and MHF. Therefore, the four samples show peaks around 14.70°, 14.80°, 16.30°, and 22.5°. Different crystal faces have distinct properties, such as hydrophilicity, hydrogen bonding, and interfacial interactions. There are four possible crystal faces for cellulose: (100), (110), $(\bar{11}0)$, and (010) [40]. The calculated crystallinity index (CI), based on the peak intensity ratio, shows that the crystallinities for OHF, MHF, THF, and HP are 59.65%, 89.55%, 79.37%, and 78.81%, respectively. MHF exhibits the highest crystallinity. Higher CI values indicate a higher order and alignment of cellulose chains, which may result in greater mechanical strength, thermal stability, and biodegradability of hemp fibers [24,41]. It can be deduced that hydrous ions first attacked the cellulose molecules on their surface before penetrating the more easily reactive amorphous area. The acid’s impact worsened the hydrolysis of the amorphous region. While crystallites may also be etched, the amorphous zone is more affected. Therefore, hydrolysis of the undeveloped area often improves crystallinity [42,43]. On the other hand, crushing rollers can produce cleaner fibers when hemp is spun at more than 400 rpm, resulting in finer particles of whole bran [44]. Using high-speed (25,000 r/min) crushing and acid treatment reduced the crystallinity of HP. The crusher reduced the crystallite size, significantly reducing the crystalline fraction. As the hemp fiber was crushed, combined pressures of shear and compression mostly destroyed shorter crystalline structures, leading to a finer product. Therefore, it can be assumed that the high-speed crushing process primarily reduces fiber crystallinity.

### 4.1. Powder Particle Size and Surface Area

The particle size distribution of the hemp powder was measured, and a log-normal distribution function was fitted to the particle size histogram. The equation used for this fitting is
D = 12πσDexp (−ln 2(D/D0)2σD2)D = \frac{1}{\sqrt{2\pi \sigma_D}} \exp\left(-\frac{\ln^2(D/D_0)}{2\sigma_D^2}\right)D = 2πσD1exp(−2σD2ln2(D/D0))
where σD is the standard deviation and D is the mean particle size. (Figure 6a) shows a typical log-normal distribution fitting for the annealed sample [45]. After manually calculating the particle sizes using SEM images, the average particle size was found to be 38.68 μm, with a standard deviation of 1.85 μm. The crushing process using the XL-06B took 3 to 5 min, during which friction and shearing forces were randomly applied to the powder due to the high speed and pressure. Figure 6b shows that the powder surfaces were uneven and rough. Most powder particles had a triangular shape and sizes of roughly 0%, 2.63%, 5.26%, 28.90%, and 63.2%, respectively.

### 4.2. Composition Characterization of Hemp Powder and Fiber

The FTIR spectrum in Figure 5b was normalized for better analysis of the chemical modification of hemp fiber and powder. All four curves presented similar peak positions, indicating that, after acid treatment, temperature changes, and crushing, the characteristic group changes occurred primarily at 933 cm^−1^, 1504 cm^−1^, and 3734 cm^−1^. The main components of hemp fiber are cellulose and hemicellulose. These components correspond to the C–O, C=O, and hydroxyl groups (-OH), which form strong bonds. During hydrolysis, the β-1,4-glycosidic link is broken, though this has little impact on the strength of the C–O and O–H bonds [9]. Lignin typically exhibits absorption bands associated with C–H stretching and aromatic skeleton vibrations, while hemicellulose shows bands linked to the stretching of acetyl groups or other functional groups along their C–O bonds. At approximately 1645 cm^−1^, pectin exhibits noticeable antisymmetric stretching vibration of the carboxylate group. High-crystalline cellulose is not easily hydrolyzed, while hemicellulose has a low-energy, random amorphous structure [46]. MHF and THF show slight changes in their curves. OHF indicates that the chemical composition and temperature of the hemp fiber changed after modification. The reaction is the hydrolysis of cellulose in hemp fiber, breaking the glycosidic bonds between glucose units and producing nanocellulose, a nanomaterial with improved properties [35]. Sulfuric acid cleaves the amorphous portions of cellulose fibers during this process, producing nanocellulose with a crystalline structure. The intensity of the peaks is significantly reduced after modification.

## 5. Conclusions

This study has demonstrated the significant potential for environmental applications in the preparation and performance of ultrafine powder made from industrial hemp degumming residue. The process of converting industrial hemp residue into ultrafine powder reduces environmental pollutants while simultaneously providing an environmentally friendly method for utilizing agricultural waste. Sulfuric acid was used to modify the residue of hemp cellulose, along with a YK agent, to aid in the dilution of the chemicals into the fiber. Using SEM, AFM, and XRD imaging, the hemp fiber’s surface was observed before and after treatment. Following treatment, the fiber surface became rougher, and, after heating to 200 °C, it became significantly harsher. According to FTIR analysis, the modified fiber’s strength decreased substantially, making it easier to crush than the original untreated hemp fiber. Crushing the modified hemp fiber into powder increased its crystallinity, though it decreased after further pulverization into powder. Ultimately, the laboratory crusher produced approximately 1 kg of dry powder with a particle size of 38.68 μm in 3 to 5 min. This work presents a more efficient method for crushing hemp cellulose fibers into micro-sized dry powders. The resulting hemp ultrafine powder has a wide range of potential applications in various industrial fields, including composite materials, textiles, paper, chemicals, biology, catalysis, and medicine. This process could reduce the time and cost of producing cellulose powder while promoting the use of cellulose powder across multiple industries.

## Figures and Tables

**Figure 1 polymers-16-03473-f001:**
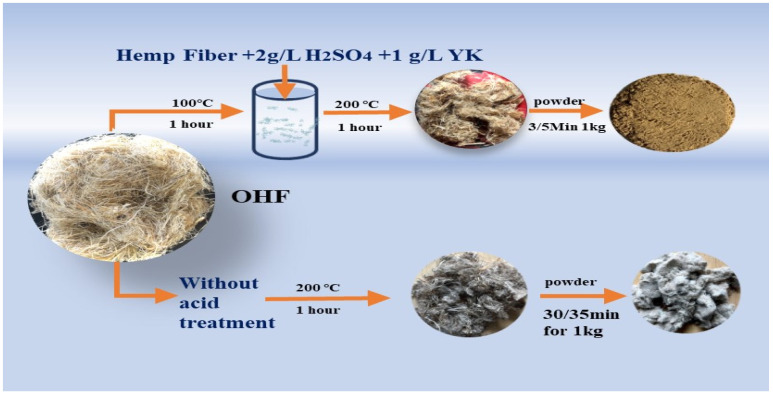
The process of making a powder without the acid treatment and with the acid treatment.

**Figure 2 polymers-16-03473-f002:**
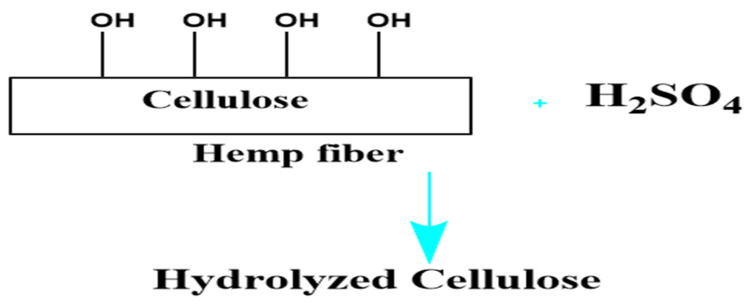
Hydrolysis reaction.

**Figure 3 polymers-16-03473-f003:**
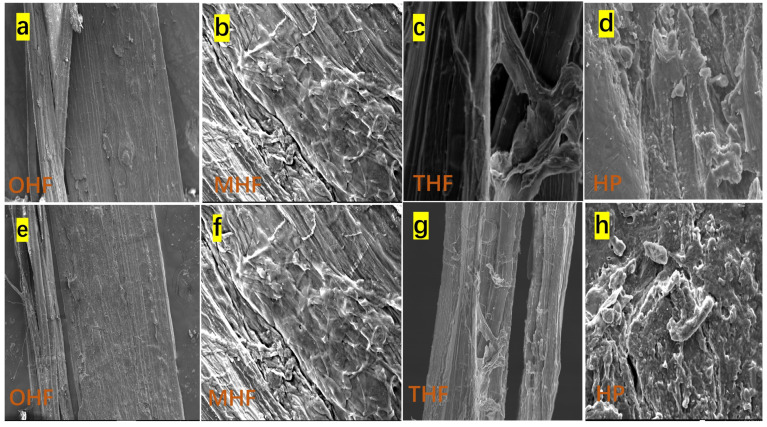
High magnification of (**a**) original hemp fiber (OHF), (**b**) modified hemp fiber (MHF), (**c**) high-temperature hemp fiber (THF), (**d**) hemp powder (HP), and low magnification. (**e**) OHF, (**f**) MHF, (**g**) THF, (**h**) HP.

**Figure 4 polymers-16-03473-f004:**
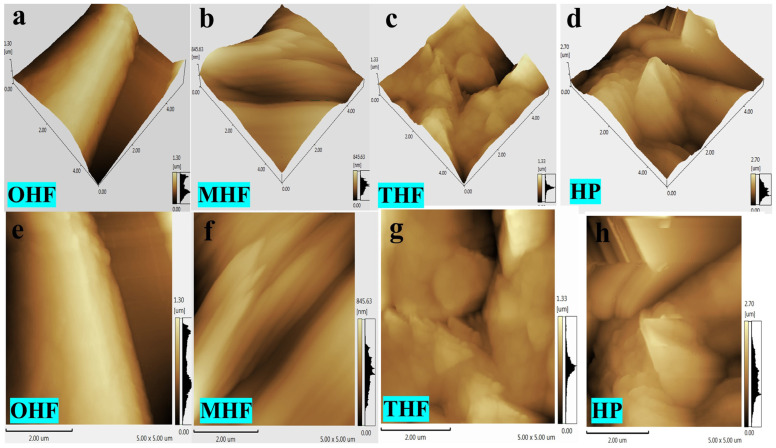
AFM scanning images measuring 5 µm × 5 μm of (**a**) original hemp fiber (OHF), (**b**) modified hemp fiber (MHF), (**c**) high-temperature hemp fiber (THF), and (**d**) hemp powder (HP). (**e**) OHF, (**f**) MHF, (**g**) THF, (**h**) HP.

**Figure 5 polymers-16-03473-f005:**
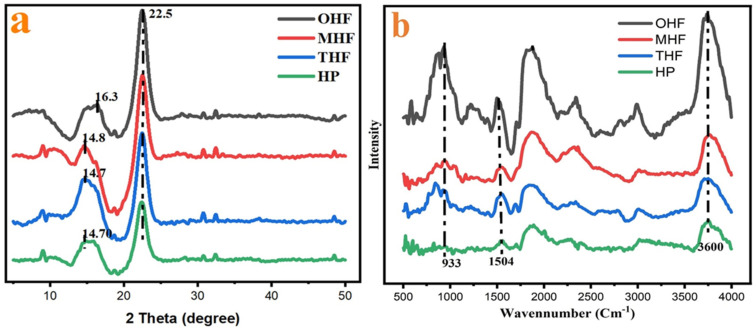
(**a**) XRD and (**b**) FTIR result of original hemp fiber (OHF), modified hemp fiber (MHF), high-temperature hemp fiber (THF), and hemp powder (HP).

**Figure 6 polymers-16-03473-f006:**
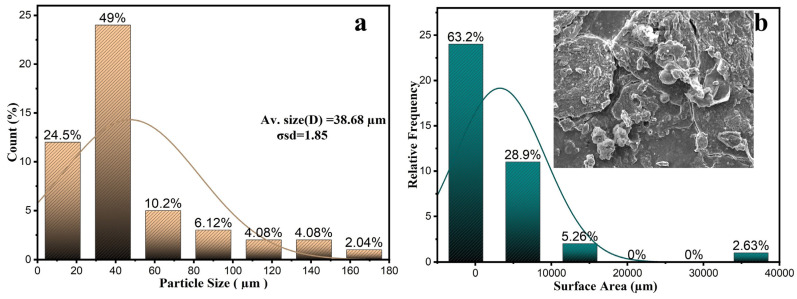
(**a**) Particle size (μm) and (**b**) surface area (μm).

## Data Availability

Data are contained within the article.

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
