# Peer review of "A Study on the Preparation and Performance of Ultrafine Powder Made of Industrial Hemp Degumming Residue"

_polymers, 2024, doi:10.3390/polym16243473_

Round 1

Reviewer 1 Report

Comments and Suggestions for Authors

Title: The physicochemical method uses industrial leftover hemp fibre cellulose to make micron-sized hemp powder

Manuscript Number: polymers-2817048

Comments:

This study developed a method for recycling industrial hemp waste into ultrafine powder. The study incorporates various analytical techniques such as AFM, SEM, and XRD to investigate the changes in surface roughness, fiber morphology, and crystallinity. The abstract is well written including all the elements of a good abstract. The authors provide a detailed background on hemp production and waste and properly highlight the significance of recycling hemp fiber. The method section contains useful information but can be improved.

The study is relevant and timely for scientific and environmental reasons. However, the authors need to address the following concerns before the manuscript can be published:

1.      The overall language should be improved tremendously. The author should write in one format, either past tense or present tense but not both. In most parts of the manuscript, but especially section 2.2, the authors report the methodology using a mix of past or present tense and instructional sentences. There are many instances of incomplete sentences. For example, line 84 is both instructional and incomplete. This is just one example of many appearing throughout the manuscript, especially in the conclusion section. Also, there are instances of double words such as “were were” in line 86. The authors should proofread the entire manuscript to address these issues and consider improving the sentence structures to enhance readability.

2.      The conclusion needs to be rewritten considering issue 1 above to enhance coherence and the reader’s comprehension.  

3.      The authors should correct the chemical formulas such as C2H6O to C2H6O. Also, the units should be modified such as gm? g?. Also, please unify the use of ultra-fine or ultrafine.

4.      Make sure to define abbreviations at their first mention such as OHF, MHF, THF, and HP.

5.      In line 80, there is an occurrence of a different font size.

6.      The authors have great images. However, unify the labelling such as (a), and (b). In Figure 8, replace it with one with the correct image size to avoid stretching. This could tremendously improve the overall quality of the article.

Comments on the Quality of English Language

The content of the manuscript is interesting to the scientific community but extensive editing of the English language is required. I have addressed this in my review. If this is addressed, it will tremendously improve the quality of the paper, enhance readability and promote the reader's comprehension.

Author Response

Point-1:A beaker containing 200 ml of boiled deionized water was placed into a 100°C water bath for 1 hour. Sequentially, 2 g/L sulfuric acid raw material, 1 g/L YK agent, and 10 g of leftover untreated hemp fiber (cut to 2 mm × 3 mm) were added to the beaker. The bath ratio was 1:20.
After heating the beaker for 1 hour, the hemp fiber was washed 7–8 times to remove excess foam caused by the YK agent. The pH value was 7.
Excess water was removed using a padding machine or hand roller.
Finally, the hemp fiber was placed in an oven and baked at 200°C for 1 hour.
The powder was prepared for AFM by dissolving it in C2H6O at 39.82 g/L.
Hemp powder was added at a concentration of 0.70 g/L. The dissolution process took 30 minutes.
The acid modification process is illustrated in Fig. 3(b).

Reviewer 2 Report

Comments and Suggestions for Authors

Dear Authors,

The manuscript of Sarker Md Shamim et al., entitled “ The Physicochemical Method Uses Industrial Leftover Hemp Fibre Cellulose to Make Micron-Sized Hemp Powder is presenting only another method for converting the leftover hemp fiber into micron-sized powder.

First of all, in my opinion, the title is not well-worded; the introduction doesn't provide any up to date information on the topic and the need for this study at all.

The text of this manuscript is not concise and clear, and often it is not understood what the authors want to say.

Bibliographic source 3 is quoted before 2.

I do not understand the numbering… there is only one figure 2 a and 3 a

SEM images should be compared to the same magnification.

Figure 7 is missing.

I highly advise rewriting and resubmitting the current manuscript!

Comments on the Quality of English Language

The quality of English language must be improved!

Author Response

Thank you for pointing this out. I agree with this comment. Therefore, I have changed the point. And now I'm going to upload a new manuscript. 

Reviewer 3 Report

Comments and Suggestions for Authors

The experimental article “The physicochemical method uses industrial leftover hemp fiber cellulose to make micron-sized hemp powder” is devoted to grinding hemp fiber into an ultra-fine powder. The article is very short in length and does not reveal the novelty of the pattern of the process under study and the practical significance of the result obtained by the authors. Directly in the introduction, the authors do not describe the method for producing short fiber, which they intend to turn into powder, and it is also not clear what this powder is needed for. Considering the rating of the Polymers publication, the manuscript requires serious revision, since a large number of questions arise for the authors, as well as a number of comments and recommendations, which are formulated in a list. It is possible that authors need to carefully study the rules of this publication, including from the names of sections to the rules for formatting references.

Notes and recommendations:

1. It is recommended to change the name to reflect the scientific novelty of the results obtained, such as “...Ultrafine hemp cellulose powder...”.

2. The abstract must be rewritten indicating the novelty of the pattern of the process under study and the practical significance of the result obtained by the authors. Correct a sentence that makes no scientific sense: “Particle size, crystallinity, fiber surface, and strength increased and decreased.”

3. In the first sentence of the article, the authors lost the name of the type of raw material being discussed: Worldwide production indicates a range of 32,140 ha to 142,883 tonnes, resulting in an average yield of 4.45 tonnes ha-1 [1].

4. In the first paragraph of the introduction, it is recommended to clearly define the type of waste that this article is devoted to. It is very difficult to understand that we are talking about short hemp fiber, the share of which is only 7.7% (line 38). Here it is necessary to give examples of the use of short hemp fiber and objectively criticize these examples.

5. Fundamentally revise introduction links to include powerful hemp fiber reviews with release dates 2020-2023.

6. Lines 46-68: It is important to rewrite this paragraph with a clear explanation of the need to obtain ultrafine powders from short hemp fibers.

7. Once the introduction has been corrected, authors should state the purpose of their research. As presented, it is absolutely not clear why short hemp fiber needs to be turned into ultra-fine powder, and in the annotation it is written that “alkali” was used, in line 70 - “acid”, why these substances differ from “toxic and complex chemicals” ( line 63).

8. It is necessary to provide the component composition of the original fiber and reveal the chemical essence of the fiber modification process described in section 2.2. Acid Treatment and Modification of hemp Fiber. What is the substance “YK agent”? Do the authors guarantee its complete removal during the washing process “after modification”?

9. Throughout the text, it is necessary to determine the terminology: what did the authors work with? With fiber or with cellulose? This is extremely important for the Polymers publication.

10. What is new about the process of crushing fiber in a crusher? There is no point in presenting Figure 1 in the article, since it does not reflect the essence of the process itself.

11. Figure 2. It should be completely removed, since in the introduction there is no statement of the goal of obtaining nanocellulose by acid hydrolysis of cellulose, in the “Materials and Methods” there are no methods for determining the nanoscale size of cellulose.

12. Figure 3 should be deleted. In Saplimentari you can provide normal photos of the appearance of the fiber and powders obtained from it.

13. Lines 163-165. It is necessary to provide evidence of the dissolution of “lignin, hemicellulose, and pectin, among other non-cellulosic fiber components” by sulfuric acid, as well as the oxidation of cellulose in hemp fiber during the experiment.

14. Expressions such as “cellulose surface's” are not acceptable for the Polymers publication.

15. Line 185. The authors must decide how the fiber was treated: “After the alkali treatment...”.

16. Line 194. Authors must prove the fact of obtaining nanocellulose, and not copy this fact from other people’s publications.

17. It is necessary to bring the material balance of the process of converting “fiber into nanocellulose” to section 4. Crystallinity of Hemp Fiber and Powder. And in section 4, comment on changes in the CI value in the pattern of this material balance.

18. Figure 6. The authors should show on the FTIR spectrum of the original fiber the characteristic frequencies for lignin, hemicelluloses and pectin, which “dissolve” when treated with acid.

19. Authors note Figure 7(a).

20. In section 4.2. Composition Characterization of the Hemp Powder and Fiber lacks the data promised by the title. The authors change their point of view regarding lignin in fiber. This section should be the first in the Results, and the description is given not on the basis of FTIR spectra, but on the actually determined component composition of the original fiber and the product of its modification. As it turns out by the end of the short manuscript, there is no nanocellulose.

21. Taking into account the above, the conclusions must be redone fundamentally. The first two sentences in the Conclusions make no sense.

Author Response

Comments 1: It is recommended to change the name to reflect the scientific novelty of the results obtained, such as “...Ultrafine hemp cellulose powder...”.

Comments 2:The abstract must be rewritten indicating the novelty of the pattern of the process under study and the practical significance of the result obtained by the authors. Correct a sentence that makes no scientific sense: “Particle size, crystallinity, fiber surface, and strength increased and decreased.”

Comments 3: In the article's first sentence, the authors lost the name of the type of raw material being discussed: Worldwide production indicates a range of 32,140 ha to 142,883 tonnes, resulting in an average yield of 4.45 tonnes ha-1 [1].

Comments 4: In the first paragraph of the introduction, it is recommended to clearly define the type of waste that this article is devoted to. It is very difficult to understand that we are talking about short hemp fiber, the share of which is only 7.7% (line 38). Here it is necessary to give examples of the use of short hemp fiber and objectively criticize these examples.

Comments: 5. Fundamentally revise introduction links to include powerful hemp fiber reviews with release dates 2020-2023.

Comments 6: Lines 46-68: It is important to rewrite this paragraph clearly explaining the need to obtain ultrafine powders from short hemp fibers.

Comments 7. Once the introduction has been corrected, authors should state the purpose of their research. As presented, it is not clear why short hemp fiber needs to be turned into ultra-fine powder, and in the annotation, it is written that “alkali” was used, in line 70 - “acid”, why these substances differ from “toxic and complex chemicals” ( line 63).

Comments 8. It is necessary to provide the component composition of the original fiber and reveal the chemical essence of the fiber modification process described in section 2.2. Acid Treatment and Modification of hemp Fiber. What is the substance “YK agent”? Do the authors guarantee its complete removal during the washing process “after modification”?

Comments 9: Throughout the text, it is necessary to determine the terminology: what did the authors work with? With fiber or with cellulose? This is extremely important for the Polymers publication.

Comments 10. What is new about the process of crushing fiber in a crusher? There is no point in presenting Figure 1 in the article, since it does not reflect the essence of the process itself.

Comments 11. Figure 2. It should be completely removed, since in the introduction there is no statement of the goal of obtaining nanocellulose by acid hydrolysis of cellulose, in the “Materials and Methods” there are no methods for determining the nanoscale size of cellulose.

Comments 12. Figure 3 should be deleted. In Saplimentari you can provide normal photos of the appearance of the fiber and powders obtained from it.

Comments 13. Lines 163-165. It is necessary to provide evidence of the dissolution of “lignin, hemicellulose, and pectin, among other non-cellulosic fiber components” by sulfuric acid, as well as the oxidation of cellulose in hemp fiber during the experiment.

Comments 14. Expressions such as “cellulose surface's” are not acceptable for the Polymers publication.

Comments14. Expressions such as “cellulose surface's” are not acceptable for the Polymers publication.

Comments 15. Line 185. The authors must decide how the fiber was treated: “After the alkali treatment...”.

Comments 16. Line 194. Authors must prove the fact of obtaining nanocellulose, and not copy this fact from other people’s publications.

Comments 17. It is necessary to bring the material balance of the process of converting “fiber into nanocellulose” to section 4. Crystallinity of Hemp Fiber and Powder. And in section 4, comment on changes in the CI value in the pattern of this material balance.

Comments18. Figure 6. The authors should show on the FTIR spectrum of the original fiber the characteristic frequencies for lignin, hemicelluloses and pectin, which “dissolve” when treated with acid.

Comments19. Authors note Figure 7(a).

Comments 20. In section 4.2. Composition Characterization of the Hemp Powder and Fiber lacks the data promised by the title. The authors change their point of view regarding lignin in fiber. This section should be the first in the Results, and the description is given not based on FTIR spectra, but on the actually determined component composition of the original fiber and the product of its modification. As it turns out by the end of the short manuscript, there is no nanocellulose.

Comments 21. Taking into account the above, the conclusions must be redone fundamentally. The first two sentences in the Conclusions make no sense.

 ***********************************************************************************************

Response 1,: Thank you for pointing this out. We agree with this comment. Therefore, we have…It was simple and effective for converting the leftover hemp fibre into ultrafine powder. In about 3 to 5 minutes, approximately 1 kg of dry ultrafine powder with a particle size of 38.68μm was produced. [abstract, paragraph, and lines 20-22]

Response 2:- Thank you for pointing this out. We agree with this comment. Therefore, we have…The decrease in intensity, fiber surface, degree of crystallinity and grain size was analysed. It became apparent that fibre strength decreased, and fibre roughness significantly increased after acid treatment. The degree of crystallinity of the broken fibers decreased significantly. [abstra, paragraph, and lines 17-19]

Response 3: Thank you for pointing this out. We agree with this comment. Therefore, we have…Worldwide hemp production indicates a range of 32,140 ha to 142,883 tonnes, resulting in an average yield of 4.45 tonnes ha-1 [1]  ( page-1, 28-29)

Response 4: Thank you for pointing this out. We agree with this comment. Therefore, we have Biomass, stalks, and fibrous material are the main waste types connected to industrial hemp degumming and processing. Hemp fiber has a short fiber of 7.72% [6], hemp stalks 40% and yields 60 % waste. Moreover, the rate of total waste from opening to spinning can range from 8.5% to 18% . [page-1 and lines 37-39]

Response 5,6: Thank you for pointing this out. We agree with this comment. Therefore, we have…Hemp waste can be recycled in several ways, the two main types being chemical and physical procedures. Particle technology has shown that turning materials into ultrafine powders affects their energy, activity, and surface area since the middle of the 20th century[10]. The creation of specialized processing equipment has been prompted by these modifications. It became apparent that the materials displayed certain modifications after being processed into an ultrafine powder, including surface area, surface energy, activity, and interface characteristics [11, 12]. Possessing an in-depth knowledge of ultrafine particles and their properties, ultrafine powder has been the focus of processing equipment development. Because of its altered surface characteristics, ultrafine hemp powder is especially intriguing since it can stabilize formulations found in everything from food to cosmetics.The process is possible to modify hemp ultrafine powder to have balanced surface amphiphilicity, which makes it capable of stabilizing water in oil emulsions, such as those found in food, cosmetics, and clothing [13]. It is widely utilized in various industrial fields, including composite materials, chemicals, biology, catalysis, and medicine, demonstrating extensive applications [14, 15]. Hemp ultrafine powder is used for reinforcing composites; hemp fibers have several advantages over synthetic fibers and can be effectively employed in many applications [16], such as biofuel, agricultural, animal bedding, industrial and automotive, bridgeable plastics, textile, paper, construction material, food, beverage, cosmetic and personal care products. Hemp fibres are versatile and have various advantages over synthetic fibers when used to improve composites. Hemp fibre mostly consists of cellulose, hemicellulose, lignin, and waxy materials [17]. The macromolecular polysaccharide hemp cellulose comprises β-1, 4 glycosidic linkages, and D-glucose. It is a type of covalent bond. Three components contribute to the high chemical reactivity of cellulose, a polysaccharide hydroxyl group – OH [2]. Because of its hydroxyl groups, cellulose is a polysaccharide with a high degree of chemical reactivity. However, cellulose's strong chemical connections, orientation, crystallinity, and high polymerization levels make it difficult to turn hemp fiber leftovers into ultrafine powders. According to their chemical solid bonds, orientation, crystallinity, and high levels of polymerization, producing ultrafine powders from residue hemp fibers is difficult. There are several ways to make cellulose powders: mechanical and chemical [18]. Mechanical or chemical techniques are used to create cellulose powders. To obtain fine particles, the chemical technique frequently uses complex and dangerous compounds [19, 20]. In comparison, the mechanical crushing method is easier, but because of the great flexibility and strength of the fibers, the pulverization process is more challenging and ineffective [21]. Hemp fiber is very strong and durable, with a tensile strength of about 550 MPa it’s higher than the tensile strength of cotton, which is 287MPa, and wool, which is 100 MPa [22].A few research investigations have been made on the mechanical and chemical milling of wool and silk powder [15, 23]. The cotton and wool powder with a particle size of around 60μm and 40μm were produced using freeze-milling and millstone milling techniques. The processing of the fibre into dry powder for 1 kilogram took approximately 1.5 hours[9, 18]. ( Page-2, Line 46-68)

Response 7: Thank you for pointing this out. We agree with this comment. Therefore, we have…Either chemical or mechanical processes are used to create cellulose ultrafine particles. To create fine particles, the chemical method frequently uses dangerous and complicated compounds. [19, 20] In this work, hemp fibres were first treated with acid to affect their morphology and make it easier for them to break into an effective powder. ( ‘page-2 line -63,70)

Response 8: Then, 2 g/L sulfuric acid raw material, 1 g/L YK agent,The YK agents aid in the dilution of chemicals into fibers ( page-2 line 85)

Response 9,: Thank you for pointing this out. We agree with this comment. Therefore, we have…Firstly, hemp fiber was cut from 2 mm to 3 mm, and small scissors were used before chemical treatment. About 1 kg of modified hemp fiber with a short length was used in a small Chinese Medicine crusher XL-06B (Aashia Company, China). (Page-2, Line-94)

Response 10: Thank you for pointing this out. We agree with this comment. Therefore, we have removed the crusher Figure.

Response 11: Thank you for pointing this out. We agree with this comment. Therefore, I agree with you I have changed the structure.

Response 12: Thank you for pointing this out. We agree with this comment. But that structure more indicates the process as well the fiber to powder.

Response 13: Thank you for pointing this out. We agree with this comment Sulfuric acid could dissolve lignin, hemicellulose, and pectin, among other non-cellulosic fiber components. The cellulose fiber in hemp was hydrolyzed and oxidized by sulfuric acid [27].  Hemp fiber's hollow structure was exposed, and its surface etched. More cellulose molecules were exposed as the fiber's specific surface area increased when it broke down to a certain extent [27, 28]. After being treated with diluted sulfuric acid and pretreated with aluminium sulfate, the clearance rate of hemicellulose was 98.05%, High lignin removal was achieved because of specific interactions with the p-TsOH hydro trope that aided lignin removal and Pectin is frequently hydrolyzed in a variety of procedures using acid treatments [29, 30].

Response 14: Thank you for pointing this out. We agree with this comment…  Therefore, upon etching, the degree of hemp fiber surface cracking was identical, and the hemp fiber surface's protuberance height tended to be more homologous than that of OHF. On the other hand, the HP had the highest roughness (d-h) because of The High-speed operation using different kinds of blade-focused powders, which were randomly sheared and crushed in various directions [33]. It was hard to ensure uniform crushing of each point on the surface of the powder [33, 34].

Response 15: Thank you for pointing this out. We agree with this comment…This can be seen in Fig.1. After the acid treatment and high temperature, the original gray hemp fiber turns brown, and without acid, it looks like ash black. So, the acid treatment process essentially destroys the strength of the hemp fibers, making them more easily pulverized into micron-sized powder.

Response 16: Thank you for pointing this out. We agree with this comment…The breakdown of the hydrogen bonds in cellulose that is amorphous, results in the formation of highly crystalline cellulose nanocrystals [36]. The essence of using sulfuric acid to obtain nanocellulose is to break the 1-4 glycosidic bonds between cellulose macromolecules and remove the amorphous regions of cellulose [37] ( Page-6 Line-194 )

Response 17: Thank you for pointing this out. We agree with this comment…

Response 18: Thank you for pointing this out. We agree with this comment…Lignin usually exhibits absorption bands associated with C-H stretching and aromatic skeleton vibrations. Hemicelluloses exhibit bands linked to the stretching of acetyl groups or other functional groups along their C-O bonds. Around 1645 cm-11, pectin exhibits a noticeable antisymmetric stretching vibration of the carboxylate group

Response 19: Thank you for pointing this out. We agree with this comment.

Response 20: Sulfuric acid cleaves the amorphous portions of cellulose fibers during this process, producing nanocellulose, a crystalline structure. Certain peaks that were present in the raw fibers will no longer be present in the FTIR analysis of nanocellulose, indicating the removal of non-cellulosic components like lignin and hemicellulose.

Response 21: According to research, there is a lot of potential for environmental applications in the preparation and performance of ultrafine powder generated from industrial hemp degumming residue. The process of converting industrial hemp residue into ultrafine powder lowers pollutants in the environment while simultaneously providing an environmentally friendly method to use agricultural trash. The sulfuric acid was modified from the residue hemp cellulose.

Round 2

Reviewer 2 Report

Comments and Suggestions for Authors

Dear Authors,

In this revised version of the manuscript there is still only one Figure 2 a and Figure 3 a…please make the correction. The positioning of figures should be done in text as soon as they are referenced.

Figure 2b - the reaction from the acid pretreatment of cellulose in hemp should be rewritten to receive a more academic form.

The English language of the manuscript must undergo extensive editing.

Therefore, I suggest that the manuscript undergo a thorough correction process.

Comments on the Quality of English Language

The English language of the manuscript must undergo extensive editing.

Author Response

In this revised version of the manuscript there is still only one Figure 2 a and Figure 3 a…please make the correction. The positioning of figures should be done in text as soon as they are referenced.

Response 1: Thank you for pointing this out. We agree with this comment. Therefore, I have changed Figures 2a and 3a instead of Figures 1(a) and 2 (b) as I submitted my revised manuscript.

Figure 2b - the reaction from the acid pretreatment of cellulose in hemp should be rewritten to receive a more academic form.

Response 2: Thank you for pointing this out. We agree with this comment As Figure 2 (b) Hydrolyzed reaction I have changed the chemical structure as I have submitted  manuscript

The English language of the manuscript must undergo extensive editing.

Response 3: Thank you for pointing this out. We agree with this comment. I have changed and submitted new revised manuscript.

Reviewer 3 Report

Comments and Suggestions for Authors

The experimental article under the new title “Study on the Preparation and performance of ul-trafine powder made of industrial hemp degumming residue” has undergone major changes: the authors figured out the name of the “raw material”, the reagent used in their process, inserts in the text and The list of references updated the presented results. But the article is not ready for publication in such a highly rated publication. Notes for correction are given in a list.

List of notes:

1. It is recommended to correct the sentence “One of the most widely available and extensively produced varieties of industrial hemp, which generates a substantial amount of consumer-increasing waste in the form of hemp cellulose.” in the annotation, from which it follows that cellulose is a waste product in hemp processing. This statement is incorrect.

2. When describing the component composition of the fiber (even in the introduction), it must be clearly stated that “fiber” is not equivalent to “cellulose”.

3. Authors should describe the role of sulfuric acid in the treatment process. From Figure 2c it follows that the authors are not familiar with the chemical hydrolysis of cellulose, so the figure should be removed. Readers of Polymers are aware of the products that sulfuric acid hydrolysis of cellulose leads to.

4. It is necessary to provide a material balance of the entire process and indicate the yield of “powder” in terms of fine fiber, as well as in terms of the direct raw material “initial hemp”. These data do not reduce the contribution of the authors, but allow us to assess the feasibility of the process.

5. Figure 4. It is recommended to improve the quality or move it to Saplimentari.

6. Check that the links are formatted correctly. For example, reference 36 is missing the year of publication.

Author Response

  1. It is recommended to correct the sentence “One of the most widely available and extensively produced varieties of industrial hemp, which generates a substantial amount of consumer-increasing waste in the form of hemp cellulose.” in the annotation, from which it follows that cellulose is a waste product in hemp processing. This statement is incorrect.

 Response 1 : (Industrial hemp, one of the most widely available and extensively produced varieties generates a substantial amount of waste in the form of hemp cellulose. This study uses a recycling method combining crushing and acid treatment to convert leftover hemp fiber into ultrafine powder) Thank you sir  for pointing this out. We agree with this comment. Therefore, we have changed our new manuscript that page number 1—abstract first lile.

  1. When describing the component composition of the fiber (even in the introduction), it must be clearly stated that “fiber” is not equivalent to “cellulose”.

 Response 2: Thank you, sir, for pointing this out. We agree with this comment, and therefore, we have changed our new manuscript.

3.Authors should describe the role of sulfuric acid in the treatment process. From Figure 2c it follows that the authors are not familiar with the chemical hydrolysis of cellulose, so the figure should be removed. Readers of Polymers are aware of the products that sulfuric acid hydrolysis of cellulose leads to.

Response 3: Yes Sir. Thank you, for pointing this out. We agree with this comment.  And Figure 2c I have removed from the new manuscript and instated Figure 2.(b) Therefore, we have changed our new manuscript. Page number 5

  1. It is necessary to provide a material balance of the entire process and indicate the yield of “powder” in terms of fine fiber, as well as in terms of the direct raw material “initial hemp”. These data do not reduce the contribution of the authors, but allow us to assess the feasibility of the process.

Response 4:  ( Waste hemp fibres were efficiently converted into micron-sized powders by combining chemical and physical processes without complex procedures. Ultimately, after 3 to 5 minutes, about 1 kg of dry powder with a particle size of 38.68 μm was produced. This method's dry ultrafine hemp powder output and efficiency were significantly higher than ball milling, cutting milling, bead milling, millstone, and spray drying techniques. This method holds much potential for future production utilization. )Thank you sir  for pointing this out. We agree with this comment. Therefore, we have changed our new manuscript.

  1. Figure 4. It is recommended to improve the quality or move it to Saplimentari.

Response 4 : Thank you, for pointing this out. We agree with this comment and I have submitted new revised manuscript.

  1. Check that the links are formatted correctly. For example, reference 36 is missing the year of publication.

Response 6: (36.  Y. Liu, Y. Wei, Y. He, Y. Qian, C. Wang, and G. J. A. o. Chen, "Large-Scale Preparation of Carboxylated Cellulose Nanocrystals and Their Application for Stabilizing Pickering Emulsions," vol. 8, no. 17, pp. 15114-15123, 2023) Thank you, sir, for pointing this out. I  agree with this comment and I have added my  new revised manuscript.

Round 3

Reviewer 3 Report

Comments and Suggestions for Authors

The experimental article "Study on the Preparation and Performance of Ultrafine Powder Made of Industrial Hemp Degumming Residue" has been in the "major" state for a long time. The authors agreed with the comments and assure the reviewer that everything has been corrected. But in the version provided, I see the same errors, for example, Figure 4 remains unchanged. Please familiarize me with the version for the third round.

In the meantime, I am forced to make the major decision again.